# J Proteins Counteract Amyloid Propagation and Toxicity in Yeast

**DOI:** 10.3390/biology11091292

**Published:** 2022-08-30

**Authors:** Daniel C. Masison, Michael Reidy, Jyotsna Kumar

**Affiliations:** 1Laboratory of Biochemistry and Genetics, National Institute of Diabetes and Digestive and Kidney Diseases, National Institutes of Health, Bethesda, MD 20892, USA; 2Department of Chemistry, University of Connecticut, Storrs, CT 06269, USA

**Keywords:** J proteins, amyloid, polyglutamine, prions

## Abstract

**Simple Summary:**

Dozens of diseases are associated with misfolded proteins that accumulate in highly ordered fibrous aggregates called amyloids. Protein quality control (PQC) factors keep cells healthy by helping maintain the integrity of the cell’s proteins and physiological processes. Yeast has been used widely for years to study how amyloids cause toxicity to cells and how PQC factors help protect cells from amyloid toxicity. The so-called J-domain proteins (JDPs) are PQC factors that are particularly effective at providing such protection. We discuss how PQC factors protect animals, human cells, and yeast from amyloid toxicity, focusing on yeast and human JDPs.

**Abstract:**

The accumulation of misfolded proteins as amyloids is associated with pathology in dozens of debilitating human disorders, including diabetes, Alzheimer’s, Parkinson’s, and Huntington’s diseases. Expressing human amyloid-forming proteins in yeast is toxic, and yeast prions that propagate as infectious amyloid forms of cellular proteins are also harmful. The yeast system, which has been useful for studying amyloids and their toxic effects, has provided much insight into how amyloids affect cells and how cells respond to them. Given that an amyloid is a protein folding problem, it is unsurprising that the factors found to counteract the propagation or toxicity of amyloids in yeast involve protein quality control. Here, we discuss such factors with an emphasis on J-domain proteins (JDPs), which are the most highly abundant and diverse regulators of Hsp70 chaperones. The anti-amyloid effects of JDPs can be direct or require interaction with Hsp70.

## 1. Introduction

Most of the many prions of yeast propagate in the cytoplasm as infectious, misfolded, insoluble amyloid forms of normal cellular proteins. An amyloid is a highly ordered filamentous aggregate composed of a single type of protein that grows when monomers of the protein join at its ends. Yeast prions and other amyloids fold into parallel, in-register beta sheets that act as templates that guide protein monomers to adopt the same conformation as they join the fiber [1,2,3,4,5,6]. Although this type of assembly produces fibers with homogeneous structures, amyloid-forming proteins produce amyloids with different physical characteristics due to variations in the extent of the amino acids that compose the beta strands or in locations of the folds of the beta-sheet template [7,8]. In vivo, these structural differences in amyloids of the same protein can manifest as differences in phenotype, referred to as “variants” of the prion, which could be due to differences in amyloid assembly or fragility or in interactions with protein quality control (PQC) factors [9,10,11,12] (see also [13]). Proteins that compose prions often have defined regions which are frequently separable from other parts of the protein and confer this amyloid-forming characteristic.

Such “prion-like” regions of amyloid disease-associated proteins in yeast and humans can be important for governing phase separation of the proteins into physiologically important and reversible biomolecular condensates, or granules. These proteins also can be associated with disease when the prion-like regions responsible for granule formation fold improperly, leading to the formation of amyloids instead [14,15]. In yeast, for example, the prion-like region of Sup35 promotes the formation of cytosolic granules in response to sudden stress. This process is reversible and helps to maintain cell health and promote recovery when stress subsides [16]. This same region of Sup35 can form a pathologic amyloid (prion) of Sup35 called [*PSI^+^*] that depletes much of Sup35 into insoluble and typically irreversible aggregates. Similarly, human TDP-43 and FUS, for example, can be associated with disease when they form insoluble amyloid-like aggregates rather than reversible condensates [17]. Protein chaperones can reduce these “off-pathway” events and associated toxicity [18].

Much accumulated evidence shows that amyloid-based yeast prions impose a burden on cell health [19]. Fitness defects caused by prions range from moderate stress to lethality [20,21,22,23,24]. Several independent PQC systems of yeast, which protect cells from a broad range of harmful protein misfolding problems, also counteract prion toxicity and the ability of prions to become established or to propagate [21,22,25,26,27,28,29,30,31,32,33,34,35,36,37,38]. Prions that arise de novo when a component of one of these systems is missing are almost entirely eliminated when it is restored, which shows that most prions that could arise are being eliminated by these cellular anti-prion activities. Defects in combinations of these systems cause additive and synergistic effects [39]. These findings illustrate the important roles that many PQC factors have in continuously guarding against the harmful properties of amyloid-based prions.

The expression of any of a variety of amyloidogenic disease-associated proteins is highly toxic to yeast [14,40,41,42,43], which makes yeast a useful system for studying the causes of toxicity. Protein chaperones engage partially folded or misfolded proteins to prevent their aggregation and to help them adopt and maintain their native conformations. Given that an amyloid is a misfolding problem, it is not surprising that elevating the expression of chaperones, including several Hsp40 family J-domain proteins (JDPs), prevents the toxicity of amyloids in yeast, humans, and other model systems [38,41,42,44,45,46]. Using yeast to understand how conserved JDPs counteract cellular toxicity surely will help inform others of their use in approaches toward therapies for amyloidoses.

Ironically, the replication of yeast prions, which is required for them to propagate stably in expanding populations, depends on chaperone machinery that is crucial for the recovery of cells from protein-denaturing stress by resolubilizing aggregated proteins [47]. This machine is driven by Hsp104 and requires assistance from the Hsp70 system, which includes JDP and nucleotide exchange factor (NEF) co-chaperones [25,48,49]. The degree to which this machinery assists prion replication is directly proportional to the extent that Hsp104 is able to drive disaggregation of stress-denatured proteins [50,51], indicating that Hsp104 acts on both types of aggregate by using the same extraction of monomers. Inopportunely, when this PQC machinery similarly recognizes and acts on an amyloid as a harmful protein aggregate, extraction of the monomers results in dividing the fibers into more numerous pieces, or “seeds”. It fails in its attempt to eliminate the amyloid, as the incorporation of monomers into the newly replicated seeds outpaces their extraction. This replication by default thus produces enough prion seeds, which can range from tens to hundreds per yeast cell, to be distributed effectively among the cells of an expanding population [52,53].

Alterations that disrupt the function of the disaggregation machinery can disrupt prion replication, causing eventual loss of the non-dividing prions as they become diluted among the increasing number of cells of a growing population. Deactivating or deleting the non-essential Hsp104 therefore “cures” the cells of all amyloid-based yeast prions that depend on it for replication. Paradoxically, overexpressing Hsp104 efficiently eliminates some but not all prions [25,27,54]. Mutations in the N‐terminal region of Hsp104 abolish its overexpression curing without altering its protein dis-aggregation activity or its ability to replicate prions when at normal levels [51]. Thus, the normal disaggregation activity of Hsp104 is not enough to support its prion‐curing function. The mechanisms underlying overexpression curing remain unresolved.

The curing of prions by elevating or deleting Hsp104 was the first demonstration that simply altering the chaperone abundance could disrupt the propagation of yeast prions and suggested that prion phenotypes were linked to a protein folding problem [25]. Since then, deleting, mutating, or altering the abundance of Hsp70 or many of its co-chaperones was found to alter the propagation of prions [12,20,27,32,46,55,56,57,58,59,60]. Some of these effects are prion-specific, showing that prions are differentially sensitive to changes in PQC functions, which might reflect differences in their requirements for specific PQC activities. The sensitivities of prions to alterations in PQC activity make them useful as tools to monitor the specific and general physiologic functions of PQC factors [61]. Some of the curing caused by altering chaperones and co-chaperones is attributable to their effects on the function of Hsp70 or the disaggregation machinery, while others are due to their acting directly on amyloids.

Other PQC factors that can influence amyloid propagation include those involved in protein degradation processes, in collecting dispersed protein aggregates, or in sequestering misfolded proteins into any of many distinct deposition sites [29,31,62,63,64,65]. Initially, some anti-prion effects of PQC factors on prions were interpreted as being caused by simple mass action effects due to altering the protein’s normal activity. However, because so much of PQC involves complex interactions among many factors in different and often overlapping PQC processes, it is difficult to know with confidence how altering any single component will affect the activities of the others. Here, we focus on the anti-amyloid properties of J proteins.

## 2. J Protein Structure and Function

J-domain proteins (Figure 1, see [66]) are Hsp70 co-chaperones defined by a conserved J domain, which interacts with Hsp70 to regulate its ATPase and substrate binding cycle. The roughly 40-kDa Hsp40 subfamily of JDPs typically encodes an adjacent glycine-phenylalanine (GF) rich region—sometimes extended by glycine and methionine (GM)—that contributes to functional specificity, followed by a C-terminal substrate binding domain (CTD) that encodes a dimerization region near its terminus. The homologous yeast Ydj1 and human DnaJA1 are the major class-A Hsp40 subfamily members. Class A Hsp40s also possess a zinc-binding region embedded in the CTD. The homologous yeast Sis1 and human DnaJB1 are the major class B subfamily members. The human class B family can be divided into two clusters, one of which includes the highly similar DnaJB6b and DnaJB8. DnaJB6b and DnaJB8 have much shorter CTDs and a serine-threonine-rich region that binds amyloids [67]. DnaJB7 is similar but localizes to the nucleus and has a longer CTD.

### 2.1. DnaJB6b Prevents Formation, Assembly, and Toxicity of Amyloids

The age of onset and severity of Huntington’s disease (HD) correlate with the extent of expansion of a polyglutamine (polyQ) stretch in the first exon of Huntingtin, with roughly 40 or more glutamines associated with the disease. The expression of disease-associated polyglutamine proteins has been widely used to identify and characterize cellular factors that counteract their toxic effects. Early studies using different model systems found that elevated expression of chaperones and co-chaperones, particularly Hsp70 and its JDP co-chaperones, suppresses the toxicity caused by polyQ [68,69,70,71,72,73]. DnaJB6b attracted particular attention when Hageman et al. identified it and DnaJB8 as potent suppressors of the aggregation and toxicity of HttQ-119 (Huntingtin exon 1 with 119 glutamines) in cultured cells, while a variety of other chaperones and co-chaperones of the Hsp70, Hsp110, DnaJA, and DnaJB families had little to no effect [74]. Although a part of the protection from polyQ aggregation and toxicity involved interaction of the JDPs with Hsp70, protective effects were mostly independent of both Hsp70 and the heat shock response.

The expression of DnaJB6b or DnaJB8 is not in itself harmful to the cells, and a nuclear localized splice variant of DnaJB6 (DnaJB6a) similarly prevented aggregation of polyQ in the nucleus [74,75]. Varying the timing of expression showed DnaJB6b and DnaJB8 do not considerably reduce the amount of pre-existing aggregated polyQ, but they do stop further aggregation. DnaJB6b and DnaJB8 similarly counteracted the toxicity and aggregation of other expanded glutamine disease proteins ataxin-3 (SCA-82Q) and androgen receptor (AR-72Q). Deletion analysis showed that the main activity of DnaJB8 required amino acids 152–232, which encode a region rich in serine and threonine (SSF-SST, or simply S/T) and a downstream element later described as the TTK-LKS motif [76].

The S/T region of DnaJB6b and DnaJB8 mediates interaction with polyQ aggregates, which blocks the formation of high molecular weight forms of polyQ [67]. It is also needed for DnaJB6b and DnaJB8 to form large polydisperse oligomers [77], linking its ability to form large oligomeric species with its protective anti-amyloid activity. The S/T region also acts as a docking site for histone deacetylases (HDACs), and deacetylation of K216 within LKS by HDAC4 is important for DnaJB6b and DnaJB8 to neutralize polyQ toxicity. Later work showed that elevating the expression of DnaJB6b delayed polyQ aggregation and disease onset in a mouse model of Huntington’s disease, and this protection required the S/T-rich region [78]. These findings demonstrate potential for the use of DnaJB6b as a therapeutic for amyloid disorders.

Experiments using purified proteins showed that sub-stoichiometric amounts of DnaJB6b strongly block the nucleation and subsequent assembly of polyQ peptides into amyloids [75,77,79], which suggests that DnaJB6b acts by binding directly to polyQ. Consistent with this idea, structural data for DnaJB6b show that it forms dimers and oligomers [80,81] with the S/T residues located in a likely substrate-binding cleft [82]. Moreover, DnaJB1, a homologous JDP with a large C-terminal substrate-binding domain, lacks an S/T region, forms dimers only, and fails to prevent polyQ aggregation and toxicity [74,77]. Expressing human DnaJB6b in the brains of mice expressing HttQ-201 delays pathology and extends the lifespan [78]. Serine and threonine residues within the S/T region are needed for DnaJB6b to bind to polyQ in vitro and in vivo. DnaJB1 is ineffective at suppressing the formation of Aβ amyloids, but DnaJB6b has similar S/T-dependent anti-amyloid effects with purified Aβ peptides [67,79,83]. These effects are mediated by the incorporation of DnaJB6b into oligomers, which prevents their growth. Thus, DnaJB6b can inhibit the nucleation and assembly of amyloids by different proteins.

### 2.2. Yeast Prions

The most studied prions of yeast are [*PSI^+^*], composed of translation termination factor Sup35, [URE3], composed of nitrogen catabolism regulator Ure2, and [*PIN^+^*] (also referred to as [*RNQ^+^*]), composed of Rnq1, a protein of unknown function rich in asparagine and glutamine. Each of these prions propagate in the amyloid form of the respective protein in the cytoplasm. The cells propagating the prions exhibit phenotypes resembling reduction or loss of the protein’s function due to depletion of the soluble form of the protein into amyloid aggregates [84]. As with all proteins with the propensity to form amyloids, these proteins can form amyloids with different structural and physical characteristics, which in yeast manifest as prion variants [7,9,10].

Hsp104 disaggregation activity depends on cooperation with Hsp70 and its Hsp40 (JDP) and NEF co-chaperones [48,49]. Differences in the effectiveness of Hsp104 for acting on amorphous protein aggregates or the highly structured amyloids of prions can be determined in large part by the JDP component of this disaggregation machinery [49,85]. Ydj1, the major yeast class A JDP, is more effective than Sis1, the major yeast class B JDP, for disaggregating aggregates caused by exposure of cells to protein-denaturing stress, whereas Sis1 is needed for Hsp104 to act on prion amyloids in a way that promotes prion replication [49,85,86,87,88,89,90,91]. [URE3] and [*PIN^+^*] are much more sensitive to the depletion or mutation of Sis1 than [*PSI^+^*] [85,86,89], suggesting they depend more strongly on the activities of Sis1, which is needed for proper functioning of the disaggregation machinery for their replication. Table 1 lists the chaperones discussed in this review as well as their effects on amyloids and their associated toxicity.

### 2.3. DnaJB6b Protects Yeast from polyQ Toxicity

In yeast, the degree of toxicity of polyQ shows the same association with the length of glutamine expansion as in metazoa, but it depends on the presence of an endogenous prion, presumably needed to enhance the nucleation of polyQ amyloids [92,93,94]. HttQ-103 (Huntingtin exon 1 with 103 glutamines) can form aggregates in the absence of yeast prions if Hsp104 is present, but it does so much less efficiently [94]. Once established, the extent of the continued propagation of polyQ aggregates correlates with the extent of Hsp104 activity, implying Hsp104 is needed for replication of the aggregates. Curiously, prion-dependent polyQ toxicity in yeast also depends on the sequences flanking the polyQ stretch, including an adjacent FLAG epitope used as a recognition site for a commercial antibody [95]. Conversely, an adjacent poly proline region reduces toxicity, an effect associated with small polyQ aggregates being collected to a peri-nuclear aggresome-like deposition site [96]. When many small, dispersed polyQ aggregates are present per cell, they often correlate with a higher toxicity than when one or two large aggregates are present [12,95,96].

As in mammals, overexpressing DnaJB6b protects yeast from polyQ toxicity [12]. DnaJB8 also protects cells but less so, and DnaJB7, which possesses a similar S/T region and larger C-terminal domain (see Figure 1), provides noticeable but even weaker protection. While protection in cultured human cells coincides with preventing polyQ aggregation, protection in yeast was first observed to coincide with DnaJB6b driving the sequestration of many small, dispersed aggregates into one large aggregate.

Many different processes collect the dispersed aggregates of misfolded proteins, and sequestration of misfolded proteins into defined deposition sites is considered a general cellular PQC strategy to protect from the harmful effects of misfolded proteins dispersed in the cytosol [63,97]. For example, Btn2 and Hsp42 protect cells by sequestering misfolded proteins into a defined nuclear deposition site and small cytosolic bodies, respectively [97]. Btn2 and Hsp42 also cooperate to cure cells of [URE3] prions by collecting them into cytoplasmic inclusions [29,31]. When co-expressed with polyQ, DnaJB6b co-localizes with the large polyQ aggregate, which resides adjacent to the vacuole [98]. Btn2 and Hsp42 co-localize with both the small cytoplasmic polyQ aggregates and the large polyQ deposit, but neither are needed for formation of the aggregates, and they do not detoxify them. Moreover, they are dispensable for DnaJB6b to sequester the smaller polyQ aggregates to the perivacuolar site and to neutralize polyQ toxicity.

The ability of DnaJB6b to protect cells from polyQ toxicity and to sequester polyQ aggregates is also independent of Bmh1, which is important for aggresome formation, and Sti1, which collects dispersed polyQ aggregates to perinuclear deposit sites [64,96,98]. Insoluble proteins are sequestered to a perivacuolar site called the IPOD, which is adjacent to Atg8-labeled phagophore assembly sites (PAS) [63,97]. Unlike the IPOD, the polyQ deposition site does not localize near the PAS, and it forms in cells with an autophagy defect. Altogether, the data suggest the site where DnaJB6b directs polyQ is a non-IPOD, autophagy-independent perivacuolar depot [98].

Further dissection of the structural requirements and mechanism of how DnaJB6b protects yeast from polyQ toxicity showed that its ability to protect cells and to sequester polyQ aggregates are separable [98]. The sequestration of polyQ aggregates by DnaJB6b is dependent on filamentous actin and interaction with Hsp70, yet this sequestration alone does not detoxify them. A DnaJB6b mutant lacking its CTD can gather the small polyQ aggregates into one large aggregate as capably as full-length DnaJB6b, but it does not protect cells. Thus, both the small and large aggregates can be fully toxic. Although in many ways sequestration is shown to be protective, this example shows that sequestration alone is not necessarily enough to neutralize the toxicity of misfolded proteins. Conversely, although DnaJB6b with a point mutation disabling its interaction with Hsp70 cannot sequester the small aggregates, it protects cells from polyQ toxicity. While the ST region alone is enough to localize to polyQ aggregates, it does not reduce toxicity. A minimal construct containing only the ST and CTD regions, however, is enough to provide protection. Apparently, the ST region interacts with the small and large polyQ aggregates, and the adjacent CTD is involved in neutralizing the toxicity of both types of aggregates.

### 2.4. Sis1 Protects from the Toxicity of polyQ and Other Aggregation-Prone Proteins

The toxicity of polyQ that depends on [*PSI^+^*] and [*PIN^+^*] prions in yeast can be caused by the depletion of essential cellular proteins involved in translation termination, PQC of the ER, or endocytosis into aggregates of polyQ or the resident prions [40,99,100,101,102]. Overexpressing Ydj1 or Sis1 was found to modulate the toxicity of highly expressed polyQ in [*PIN^+^*] yeast cells but not in cells with [*PSI^+^*] or any prions [103]. Here, elevated expression of Ydj1 increased the polyQ toxicity, and this effect was associated with the increased size of polyQ aggregates and decreased relative solubility. In contrast, Sis1 reduced the toxicity, and the protection was associated with a decrease in the size of polyQ aggregates and an increased relative solubility of polyQ. The mechanisms underlying the toxicity and protection here were not determined but were consistent with polyQ or prion aggregates binding an essential protein and overexpressed Sis1, preventing its depletion in part by reducing the amount of aggregate that binds and depletes the protein. As Sis1 binds [*PIN^+^*] prion aggregates and polyQ [104,105], its overproduction could also prevent the depletion of an essential protein by occluding sites where it binds the prion or polyQ. Sis1 also has a role in promoting the degradation of misfolded proteins by nuclear proteasomes, however, and the binding of Sis1 to polyQ aggregates can reduce the availability of Sis1 and impair proteasome functioning [105].

Subsequent work showed that elevating the expression of Sis1 can divert polyQ from insoluble aggregates to “cloud”-like condensates which titrate Hsp70 [106]. Although overexpressing polyQ does not induce a stress response, the resulting reduction in Hsp70 availability caused by elevating Sis1 leads to the activation of Hsf1 and a protective stress response. Elevating the expression of DnaJB6b similarly induces polyQ condensates and a stress response. Independent of polyQ expression, elevating the expression of either JDP can also potentiate the heat stress response, pointing to a general role for these JDPs in sensing and responding to misfolded proteins.

In addition to Huntington’s disease-related polyQ, artificially elevating the expression of prion proteins or other aggregation-prone proteins can also cause toxicity, specifically in cells propagating prions [41,46,104,107,108]. In many instances, the toxicity is connected again to the depletion of an essential protein, either related to or unrelated to the prion. Co-overexpression of Sis1 again frequently counteracts such toxicity by reducing the depletion of the essential proteins. Two examples of this follow.

Modest overexpression of Rnq1 was found to be lethal in cells propagating [*PIN^+^*] [104]. Here, the toxicity is not due to the accumulation of excess Rnq1 amyloids or the depletion of Sis1, which is not so surprising since at normal levels, Sis1 is over 50-fold more abundant than Rnq1 [86,109]. Puzzlingly, despite this large excess of Sis1 over Rnq1, co-overexpressing Sis1 counteracts this toxicity. The protection is associated with increased aggregated Rnq1 and reduced soluble Rnq1. Sis1 could not neutralize the toxicity of Rnq1 mutated at a site where Sis1 binds. This mutant Rnq1 did not form amyloids efficiently, but it did form abundant, high molecular weight aggregates with a similar size range to the Rnq1 amyloid in [*PIN^+^*] cells. These non-amyloid Rnq1 aggregates are thought to cause toxicity by binding and depleting the spindle pole component Spc42 [110], and Sis1 reduces the toxicity by diverting Rnq1 from non-amyloid aggregation into the less toxic Rnq1 amyloid [104,111].

Overproducing an aggregation-prone region of Pin4 protein can also be lethal in [*PIN^+^*] cells [112]. Pin4 is involved in cell cycle progression and causes mitotic arrest when hyperphosphorylated in response to DNA damage. Here, again, co-overexpressing Sis1 protects from this toxicity, which is linked to Pin4 hyperphosphorylation, insoluble aggregation, and co-localization with [*PIN^+^*] prion aggregates. The toxicity is also associated with compromised ubiquitin-proteasome functioning. The toxicity could be related to cell cycle arrest or to proteasome overload. All these effects are prevented by elevating the expression of Sis1, which prevents aggregation and hyperphosphorylation of Pin4, perhaps by competing with Pin4 for binding to prion aggregates.

### 2.5. JDPs Antagonize the Propagation of Amyloid-Based Yeast Prions

An effect of elevating J proteins on yeast prions was first shown by overexpressing the Ydj1 curing cells of [URE3] [27]. This curing was later shown to require a functional J domain needed for Ydj1 to interact with Hsp70 but not its substrate-binding region. Moreover, increasing the expression of its J domain alone or the J domain of Sis1 or Jjj1 is enough to cause a loss of [URE3] [89,113]. Additionally, [URE3] is highly sensitive to any alteration in Sis1 function, and increasing the expression of Sis1 counteracts curing by Ydj1 [85]. These findings all align with the conclusions that the curing does not require the direct binding of Ydj1 to the prion and is due to competition with Sis1 for interaction with the Hsp70 component of the disaggregation machinery.

Overexpressing DnaJB6b very effectively cures yeast of [URE3] and weak variants of [*PSI^+^*] prions, but it fails to cure cells of strong variants of [*PSI^+^*] and at least one variant of [*PIN^+^*] [12]. In vitro, DnaJB6b also prevents purified Ure2 from forming amyloids [12], which can propagate as [URE3] when used to infect yeast [114], and it arrests the growth of preexisting Ure2 amyloid fibers. It also prevents the formation and growth of Sup35 amyloids prepared at higher temperatures [12], which share the physical characteristics of Ure2 amyloids and form weak [*PSI^+^*] prions when used to infect yeast [7,10]. As with the effects of DnaJB6b on polyQ, all these anti-amyloid effects require the S/T region but not Hsp70 interaction, which suggests that the curing is due to direct anti-amyloid activity.

Notably, however, in agreement with its failure to cure strong [*PSI^+^*] prions, DnaJB6b does not prevent purified Sup35 from forming amyloids at lower temperatures, which produces a structurally different amyloid that forms strong [*PSI^+^*] variants when used to infect yeast. Thus, although the general and potent anti-amyloid activity of DnaJB6b extends to amyloids composed of Ure2 and Sup35, it does not recognize or act effectively on a structurally distinct amyloid variant composed of the same Sup35 protein. As DnaJB6b has shown promise as a therapeutic [78], understanding what characteristics allow variants of amyloids composed of the same protein to be sensitive to or elude the action of DnaJB6b would be important.

Sis1 is essential for both the viability of yeast and the propagation of amyloid-based yeast prions. DnaJB1, the closest human class B Hsp40 homolog of Sis1, can substitute for Sis1 to support the growth of yeast, but the related DnaJB6b cannot. Interestingly, however, class B DnaJB6b can function in place of the class A Ydj1 to support normal growth at both optimal and elevated temperatures [12]. These functions and the ability of DnaJB6b to cooperate with Hsp70 and Hsp104 to refold aggregated protein in vitro require its interaction with Hsp70 but not its S/T region. The surface-exposed TTK-LKS region within the CTD of DnaJB6b, which is dispensable for DnaJB8 to inhibit polyQ aggregation, acts as a site for interaction with non-amyloid substrates [76] and possibly contributes to this cooperative refolding.

Defective versions of DnaJB6b with myopathy-linked mutations F89I or F93L in the GF region also complement Ydj1 functions and cure [URE3] [12]. On the other hand, while Sis1 containing the GF of DnaJB6b is functional for prion propagation, the F89I or F93L alteration in this hybrid protein disrupts its ability to promote the propagation of some variants of [*PIN^+^*] prions [115]. These findings, which are consistent with earlier data demonstrating the importance of the GF region for the functional specificity of Hsp40 JDPs [109,116], show that the GF region can be more important for prion propagation than for prion curing. Together, these data highlight the JDP structure and function differences needed for cooperation with Hsp70 both to counteract toxic protein misfolding and to support important physiological processes.

Consistent with the direct action of DnaJB6b on amyloids in vivo, interaction with Hsp70 is not needed for DnaJB6b to cure prions, and curing is enhanced if this interaction is disrupted. The CTD also is important but not essential for curing [URE3] [12]. Ydj1 also cures cells of [URE3] and weak [*PSI^+^*], but it requires interaction with Hsp70 and cures them with different kinetics [12,27]. Monitoring the fluorescence of [URE3] aggregates during curing showed that Ydj1 is associated with both an increase in size and decrease in number of [URE3] prion aggregates, whereas DnaJB6b causes prion aggregates to get progressively smaller [65]. Elevating Sis1 counteracts curing by Ydj1 but not by DnaJB6b [12]. Clearly, Ydj1 and DnaJB6b cure cells of [URE3] by different mechanisms.

Ydj1 might act by helping gather prion fibers together or by preventing Sis1 from helping keep them apart. It also might act by competing with Sis1 for interaction with Hsp70 to assist the division of fibers. Monitoring how J domains alone alter processing of fluorescent aggregates during the curing of [URE3] might put constraints on these possibilities. In contrast, DnaJB6b reduces both the size and number of aggregates, presumably by acting directly to arrest prion growth, as it does with other amyloids, without interfering with the fibers’ dividing. This effect on [URE3] contrasts with its collecting dispersed polyQ aggregates into a single site, showing that the anti-amyloid activity of DnaJB6b can work in different ways in vivo. DnaJB8 also cures [URE3] but again with reduced efficiency, such as its reduced ability to protect yeast from polyQ toxicity [12].

### 2.6. Sis1 Protects from the Latent Toxicity of [PSI^+^]

When expressed in place of full-length Sis1, Sis1JGF, a C-terminally truncated Sis1 lacking its conserved substate-binding domain, supports cell growth and the propagation of prions [*PSI^+^*] and [*PIN^+^*] but not [URE3] [22,85,86,117]. The ability of Sis1JGF to support growth and to propagate prions both require interaction with Hsp70. Moreover, a point mutation in Sis1 that prevents it from interacting with Hsp70 destroys Sis1’s function in vivo. These findings demonstrate that the essential functions of Sis1 and the role of Sis1 in propagating prions do not require normal CTD-mediated substrate binding activity but rather the Sis1-specific regulation of Hsp70.

Many studies describe how the propagation of several different prions and prion variants can depend on various chaperones, particularly JDPs and even subregions of JDPs, but very few determine whether failure to support prion propagation could be linked to failure to protect from prion toxicity. In a somewhat more natural system than ectopic overexpression of aggregation-prone non-yeast proteins, in cells expressing Sis1 lacking its CTD in place of full-length Sis1, all endogenous variants of [*PSI^+^*] tested were toxic to a degree that correlated with the strength of the variant phenotype, ranging from mild to lethal [11,38,118]. Thus, while the CTD is dispensable for both cell growth and the propagation of [*PSI^+^*] variants, it is required to protect cells from the latent toxicity of all these variants [22,38]. Swapping CTDs between Sis1 and Ydj1, which swaps their ability to support the growth of yeast in place of Sis1, also swaps their ability to support prion propagation and to protect from [*PSI^+^*] prion toxicity [85,117]. The CTD is therefore a unique, necessary, and transferable part of Sis1 that confers protection from toxicity for a broad range of [*PSI^+^*] prion variants.

The toxicity of [*PSI^+^*] in cells expressing Sis1JGF is caused primarily by the depletion of Sup35 due to the reduced ability of Sis1JGF to keep the essential Sup35 soluble enough to support growth [38]. The depletion of Sup45, the release factor partner of Sup35, into the increased mass of insoluble Sup35 in [*PSI^+^*] cells expressing Sis1JGF also contributes to the toxicity. Sis1 counteracts the toxic depletion of Sup35 into prion aggregates, possibly by ensuring the monomers of Sup35 extracted from [*PSI^+^*] prions refold properly or do not rejoin the aggregates or by limiting the incorporation of preexisting soluble Sup35 into the aggregates. By performing its normal role of helping keep proteins soluble, Sis1 allows the propagation of what otherwise would be lethal [*PSI^+^*] prions.

## 3. Overview

DnaJB6b can bind several amyloid-forming proteins and disrupt their ability to form amyloids in vitro and in vivo. This activity protects human cells, animals, and yeast from amyloid-associated toxicity. In both human and yeast systems, all anti-amyloid and protective effects require the interaction of DnaJB6b with amyloids but not with Hsp70. These observations suggest that protection is mediated through a direct interaction of DnaJB6b with amyloids rather than its ability to cooperate with and regulate Hsp70. However, DnaJB6b does not inhibit the aggregation of polyQ in yeast as it does in human cells, and the way it protects yeast cells from polyQ seems to be through a mechanism different from simply disrupting the amyloid assembly pathway. It might bind and detoxify amyloid forms of polyQ that do form or divert polyQ to a non-toxic, non-amyloid aggregation pathway. The dependence of DnaJB6b on cooperation with Hsp70 for the sequestration of polyQ in yeast without necessarily detoxifying the aggregates is an intriguing observation that might be related to differences in the ways DnaJB6b interacts with PQC factors other than Hsp70 in the two systems.

When DnaJB6b acts on preexisting [URE3] prions composed of Ure2 amyloids, it causes them to become progressively smaller until the cells are cured of the prion. This curing occurs with or without the interaction of DnaJB6b with Hsp70. Here, the effect seems more likely to be due to DnaJB6b arresting the assembly of Ure2 amyloids, which would allow the extraction of monomers from the fibers by the disaggregation machinery to outpace the incorporation of monomers. Accordingly, in vitro, DnaJB6b prevents Ure2 and Sup35 from forming amyloids that produce [URE3] and weak [*PSI^+^*] prions, which broadens the range of its anti-amyloid activity. However, it shows little ability to suppress the formation of a variant of Sup35 amyloid whose physical characteristics differ from amyloids composed of the same Sup35 protein that underlies the weak [*PSI^+^*] phenotype. This difference has important implications for the use of DnaJB6 as a therapeutic. Together, these observations illustrate the breadth and limitations of how DnaJB6b acts on various amyloids both in vitro and in vivo.

Yeast has no counterpart to DnaJB6b that possesses similar anti-amyloid activity, but the related JDP Sis1 protects yeast from the toxicity caused by the overproduction of many different amyloid-forming and aggregation-prone proteins and the latent toxicity of [*PSI^+^*] prions. It accomplishes this by counteracting aggregation, driving aggregation, or impeding the toxic depletion of essential cellular proteins. Thus far, no other endogenous JDP has been found to be as broadly effective at protecting yeast from toxic protein aggregation as Sis1. Sis1 is also crucial among the yeast JDPs for directing the activity of the disaggregation machinery in a way that promotes amyloid propagation, which is necessary in most instances for the toxicity of an overproduced amyloid-forming protein. Whether the protective effects of Sis1 are attributable to how it affects this process or if Sis1 has an ability to interact with amyloids in general in a way not shared with the other JDPs has yet to be clarified.

Additional gaps in our understanding of how cells respond to amyloids include how the CTD of DnaJB6b detoxifies polyQ aggregates. DnaJB6b without its CTD can bind the smaller dispersed aggregates and sequester them into single large foci, but the CTD is needed to detoxify both small and large aggregates. Since binding alone is not enough to provide protection, DnaJB6b would seem to be unable to simply block essential cellular factors from being depleted into the aggregates by occupying the surface of polyQ agregates. Is the CTD acting directly on the polyQ aggregates or mediating its effects through interactions with the PQC factors? Alternatively, is the CTD important for promoting oligomerization in a way that makes amyloid binding more effective?

Similar details on how Sis1 reduces the incorporation of Sup35 into [*PSI^+^*] prions would help our understanding of why Sis1 is unique in its prion-protective effects. Continued study will provide the needed details to understand how these JDPs perform these important functions in cells and to what degree their ability to counteract the toxicity of similar or different amyloids overlap.

## Figures and Tables

**Figure 1 biology-11-01292-f001:**
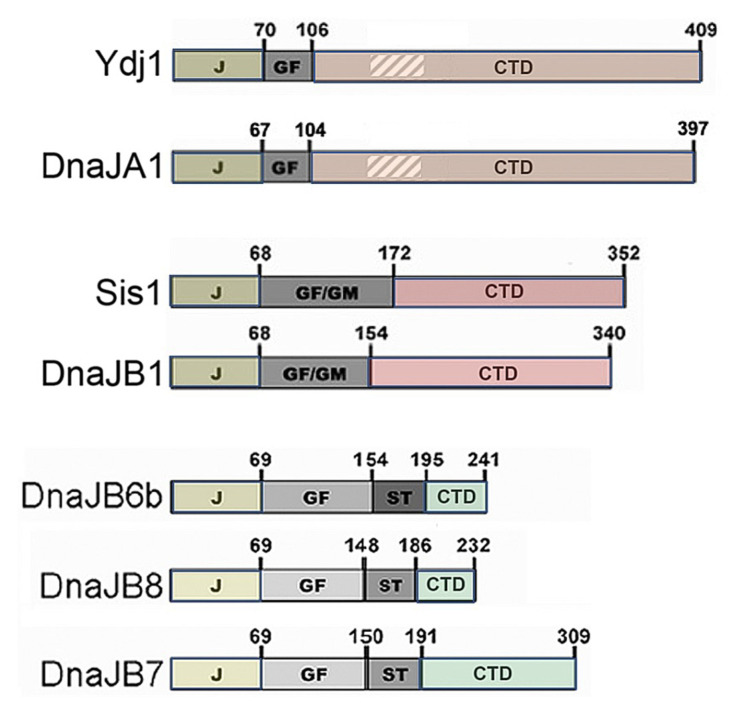
Domain structures of yeast (Ydj1 and Sis1) and related human JDPs. DnaJA1 (class A) and DnaJB1 (class B) are the closest human relatives of Ydj1 and Sis1, respectively. Numbers indicate amino acid positions at domain borders. The roughly 70 amino acid N-terminal J domains bind Hsp70, GF and GM regions confer some specificity of function, and the CTDs bind substrates. Class A JDPs Ydj1, and DnaJA1 also encode a zinc-binding region (hatched) embedded in the CTD. DnaJB6b, JB7, and JB8 comprise a class B subclass. Each has an amyloid-binding S/T region and a shorter CTD.

**Table 1 biology-11-01292-t001:** Effects of JDPs on amyloid and amyloid-associated toxicity.

JDP	Amyloid	Effects on Amyloid and Associated Toxicity ^c^
	Htt-119Q	↓ nucleation ↓ assembly ↓ toxicity in human cells, mice, and yeast
	AR-72Q	↓ nucleation ↓ assembly (androgen receptor)
	SCA-82Q	↓ nucleation ↓ assembly (spinocerebella ataxia–ataxin 3)
DnaJB6b ^a^	Aβ42	↓ nucleation ↓ assembly (amyloid-beta 42 amino acid peptide)
	Ure2	↓ nucelation ↓ assembly of Ure2 amyloid, ↓ [URE3] prions
	Sup35 ^b^	↓ nucleation ↓ assembly of Sup35 amyloid, ↓ [*PSI^+^*]^W^ prions
		No obvious effect on [*PSI^+^*]^S^ prions or associated Sup35 amyloid
	Rnq1	No obvious effect on [*PIN^+^*] (at least one prion variant)
DnaJB1	127Q	Suppresses polyQ toxicity in flies
	Prions	Needed for replication of all yeast amyloid-based prions
	Ure2	Any ↓ Sis1 function disrupts [URE3] propagation
Sis1	Sup35	↓ latent toxicity of all tested variants of [*PSI^+^*] prions
	Rnq1	↓ toxicity of ↑ Rnq1 in [*PIN^+^*] cells
	Htt-Q103	↓ toxicity and size of polyQ aggregates in [*PIN^+^*] cells
Ydj1	Ure2	Cures cells of [URE3] prions
	Htt-Q103	↑ toxicity and size of polyQ aggregates in [*PIN^+^*] cells

^a^ Or DnaJB8. ^b^ Prion variants are weak ([*PSI^+^*]^W^) and strong (*PSI^+^*]^S^). ^c^ Upward-pointing arrows indicate increase, and downward-pointing arrows indicate decrease. Unless indicated, effects in vivo are caused by elevated expression of JDP.

## Data Availability

Not applicable.

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
