# Peer review of "J Proteins Counteract Amyloid Propagation and Toxicity in Yeast"

_biology, 2022, doi:10.3390/biology11091292_

Round 1

Reviewer 1 Report

In the manuscript « J-Proteins counteract amyloid propagation and toxicity in yeast» authors discuss the role of different J-domain proteins in yeast prions propagation. The manuscript is interesting but needs to be improved.

 Summary:

Lines 12-13 “We discuss how PQC factors protect animals human cells and yeast from amyloid toxicity, focusing on how yeast and human JDP”

What authors mean in “animals human cells”?

“focusing on how yeast and human JDP” – delete “how” or add the end of the sentence

 Figure 1 – the quality is poor, the distance between different proteins should be uniform, it would be better to give the figure in color

It seems that reference list was formed automatically, but it should be corrected, for example:

Lines 493, 498 etc. - saccharomyces cerevisiae – should be in italic and capitalized

Lines 476, 490 etc. - Sup35 instead of sup35

Line 682 – comma between Alexandrov, A.I. and Ter-Avanesyan is missed

Lines 492, 514 etc. - [PSI] instead of [psi]

Line 516 – it should be [PSI(+)] and [URE3]

Line 532 – change btn2p for Btn2, [ure3] for [URE3]

Author Response

Author comments are italicized

Comments and Suggestions for Authors

In the manuscript « J-Proteins counteract amyloid propagation and toxicity in yeast» authors discuss the role of different J-domain proteins in yeast prions propagation. The manuscript is interesting but needs to be improved.

 Summary:

Lines 12-13 “We discuss how PQC factors protect animals human cells and yeast from amyloid toxicity, focusing on how yeast and human JDP”

What authors mean in “animals human cells”? To clarify that three different systems were being considered, commas were added to separate "animals, human cells, and yeast"

“focusing on how yeast and human JDP” – delete “how” or add the end of the sentence deleted "how".

 Figure 1 – the quality is poor, the distance between different proteins should be uniform, it would be better to give the figure in color Higher resolution image was sharpened and colored and incorporated.

It seems that reference list was formed automatically, but it should be corrected, for example:

Lines 493, 498 etc. - saccharomyces cerevisiae – should be in italic and capitalized fixed

Lines 476, 490 etc. - Sup35 instead of sup35 fixed

Line 682 – comma between Alexandrov, A.I. and Ter-Avanesyan is missed fixed

Lines 492, 514 etc. - [PSI] instead of [psi] fixed

Line 516 – it should be [PSI(+)] and [URE3] fixed

Line 532 – change btn2p for Btn2, [ure3] for [URE3] fixed – many other changes were made to correct the typos in the reference list

Submission Date

11 August 2022

Date of this review

17 Aug 2022 08:54:16

Reviewer 2 Report

I find this review to be a systematic analysis of the role of J-proteins in the formation, curing and toxicity of amyloids. A significant point I would like to raise is the insufficient, to my eye, discussion of the work from the Hartl laboratory, specifically the role of Sis1 in the nucleo-cytoplasmic transport of misfolded proteins, see work 10.1016/j.cell.2013.06.003 (which the authors fo reference); and 10.1038/s41467-020-20000-x (which they do not), which also highlights a signalling role of Sis1, which is not discussed in the paper. 

I suggest that these data should be asessed by the review and added into the general picture the authors present. My general impression is that the authors concentrate of the direct interaction of J-proteins with amyloids, but do not go deep into how these interactions have the observed effects, both via directly and indirect effects that involve other cellular systems. I feel that an attempt to provide such scenarios would improve the manuscript greatly.  

While in my mind the required revisions are relatively minor, I think they are quite essentail for the review to be adequately systematic in its treatment of the subject. 

Author Response

Author response is in italic below.

Comments and Suggestions for Authors

I find this review to be a systematic analysis of the role of J-proteins in the formation, curing and toxicity of amyloids. A significant point I would like to raise is the insufficient, to my eye, discussion of the work from the Hartl laboratory, specifically the role of Sis1 in the nucleo-cytoplasmic transport of misfolded proteins, see work 10.1016/j.cell.2013.06.003 (Park'13)(which the authors fo reference); and 10.1038/s41467-020-20000-x (Claips'20) (which they do not), which also highlights a signalling role of Sis1, which is not discussed in the paper. 

I suggest that these data should be asessed by the review and added into the general picture the authors present. My general impression is that the authors concentrate of the direct interaction of J-proteins with amyloids, but do not go deep into how these interactions have the observed effects, both via directly and indirect effects that involve other cellular systems. I feel that an attempt to provide such scenarios would improve the manuscript greatly.  

While in my mind the required revisions are relatively minor, I think they are quite essentail for the review to be adequately systematic in its treatment of the subject. 

Submission Date

11 August 2022

Date of this review

22 Aug 2022 11:05:09

We thank the reviewer for pointing out our oversight and we agree some discussion of that important work should be discussed. With so many studies and the many ways polyQ and prions cause toxic effects, most of our discussion has summarized relevant studies to varying degrees rather than focusing on a specific pathway, which could imply that such a pathway was the primary or only cause of toxicity. We now present a summary of the findings referred to and conclusions derived from them, here also, rather than focusing on the details and providing various scenarios. Readers interested by these findings and wishing to know more now have sources of the information to pursue their interests.